# The promotive role of lncRNA MIR205HG in proliferation, invasion, and migration of melanoma cells via the JMJD2C/ALKBH5 axis

Yujing Liu[1], Suihai Wang[2], Shanshan Wei[1], Xianwen Qiu[1], Yijie Mei[1], Lu Yan[1]*

1 Department of Dermatology, Zhujiang Hospital of Southern Medical University, Guangzhou, China,
2 School of Biotechnology, Southern Medical University, Guangzhou, China

* michelley8051@126.com

**Data Availability Statement:** All relevant data are within the paper and its Supporting Information files.

## Abstract

Melanoma is a highly malignant skin cancer. This study aimed to investigate the role of long non-coding RNA MIR205 host gene (lncRNA MIR205HG) in proliferation, invasion, and migration of melanoma cells via jumonji domain containing 2C (JMJD2C) and ALKB homolog 5 (ALKBH5). Real-time quantitative polymerase chain reaction or Western blot assay showed that MIR205HG, JMJD2C, and ALKBH5 were increased in melanoma cell lines. Cell counting kit-8, colony formation, and Transwell assays showed that silencing MIR205HG inhibited proliferation, invasion, and migration of melanoma cells. RNA immunoprecipitation, actinomycin D treatment, and chromatin immunoprecipitation showed that MIR205HG may bind to human antigen R (HuR, ELAVL1) and stabilized JMJD2C expression, and JMJD2C may increase the enrichment of H3K9me3 in the ALKBH5 promotor region to promote ALKBH5 transcription. The tumor xenograft assay based on subcutaneous injection of sh-MIR205HG-treated melanoma cells showed that silencing MIR205HG suppressed tumor growth and reduced Ki67 positive rate by inactivating the JMJD2C/ALKBH5 axis. Generally, MIR205HG facilitated proliferation, invasion, and migration of melanoma cells through HuR-mediated stabilization of JMJD2C and increasing ALKBH5 transcription by erasing H3K9me3.

## Introduction

Melanoma is an old age-related malignancy, resulting from the stochastic accumulation of inherited or acquired mutations leading to the malignant transformation of melanocytes into melanoma [1]. It is a less common but the most aggressive type of skin cancers, responsible for about 90% of skin cancer fatality [2]. Trending with increased ultraviolet exposure and cancer surveillance, the incidence of melanoma is increasing rapidly [3]. Despite the improvement of systemic therapy, the clinical outcome for advanced melanoma remains unsatisfactory [4]. Patients with high tumor burden and bone metastasis still have poor prognosis, with 5-year postoperative survival about 9.5% [5]. Given this scenario, the identification of effective

**Funding:** This study was supported in part by grants from National Natural Science Foundation of China (Grant number 82003370) and Natural Science Foundation of Guangdong Province (Grant number 2019A1515012225). The funders had no role in study design, data collection and analysis, decision to publish, or preparation of the manuscript.

**Competing interests:** The authors have declared that no competing interests exist.

therapeutic targets, especially factors targeting cell invasion and metastasis, remains in urgent need to improve clinical response of melanoma treatment.

Tumorigenesis is a complex process characterized by various epigenetic alterations, especially alterations in oncogenes and tumor suppressor genes. The ability to reprogram the epigenetic landscape has drawn enormous interests regarding the development of anti-cancer drugs [6]. Long non-coding RNAs (LncRNAs), a group of epigenetic factors with over 200 nucleotides in length, play vital roles in cancer-related cellular processes, including proliferation, migration, invasion, apoptosis, and angiogenesis [7, 8]. Especially, a number of lncRNAs regulate the growth and metastasis of melanoma cells and serve as diagnostic, prognostic, and therapeutic targets [9]. One such lncRNA, MIR205 host gene (MIR205HG), acts as a tumor booster in various cancer types, such as cervical cancer, osteosarcoma, hepatoblastoma [10–12]. Most importantly, MIR205HG knockdown has been previously demonstrated to inhibit proliferation, invasion, and migration of melanoma cells, as well as tumor growth *in vivo* [13]. However, it remains poorly understand how MIR205HG regulates melanoma cell growth.

Human antigen R (HuR, ELAVL1) is a well-known gene stabilizer that can regulate its target messenger RNA (mRNA) via an RNA recognition motif at the 3' UTR, and lncRNAs can bind to HuR, thus stabilizing the expression of downstream genes [14]. Jumonji domain containing 2C (JMJD2C), also named as histone lysine demethylases 4C, plays a role in the epigenetic regulation in tumors [15]. JMJD2C overexpression induces cisplatin resistance and expedites tumor growth in uveal melanoma [16]. The StarBase database predicted the binding of MIR205HG to HuR and HuR to JMJD2C, suggesting the potential role of MIR205HG in stabilizing JMJD2C via binding to HuR. Furthermore, JM JD2C has been demonstrated to promote the expression of ALKB homolog 5 (ALKBH5) by erasing trimethylated histone H3 Lys9 (H3K9me3) [17]. ALKBH5 is known to post-transcriptionally regulate oncogenes or tumor suppressors in an $N^6$-methyladenosine-dependent manner [18]. Significantly, ALKBH5 overexpression accelerates the metastasis of uveal melanoma by mediating epithelial-to-mesenchymal transition [19]. However, it is unclear whether MIR205HG regulates the JMJD2C/ALKBH5 axis to function in melanoma.

Taking the above associations into consideration, we hypothesized that MIR205HG plays a role in proliferation, invasion, and migration of melanoma cells via manipulation of the JMJD2C/ALKBH5 axis. The objective of our study was to investigate the molecular mechanism by which MIR205HG regulates melanoma cell growth and provide a novel theoretical reference for melanoma treatment.

## Materials and methods

### Cell culture

Human melanoma cell lines (A375, A2058, MV3, and SK-Mel-28) and human normal epidermal melanocytes (PIG1) were procured from ATCC (Manassas, VA, USA) and cultured in a Dulbecco's modified Eagle medium (Gibco; Thermo Fisher Scientific, Inc., Waltham, MA, USA) containing 10% fetal bovine serum (FBS) and placed in an incubator under the condition of 37°C and 5% $CO_2$.

### Cell treatment

All small interfering RNAs (siRNAs) targeting MIR205HG (si-MIR205HG-1, si-MIR205HG-2, and si-MIR205HG-3) and scrambled negative control (si-NC), pcDNA3.1-JMJD2C (JMJD2C), pcDNA3.1-ALKBH5 (ALKBH5), pcDNA3.1-HuR (HuR) and pcDNA3.1 empty vector were provided by Shanghai GenePharma Co., Ltd. (Shanghai, China). A375 or A2058 cells were planted in the 6-well plates at a density of $1 \times 10^4$ cells/well. When

reaching 70% confluence, Lipofectamine 3000 (Invitrogen, Carlsbad, CA, USA) was used to generate the liposome cocktail containing the aforementioned expression vectors (50 ng) or siRNA (25 nM) in Opti-MEM (Invitrogen) following the protocol. Next, cells were incubated with transfection complex for at least 8 h, and then the culture medium was replaced. After 48 h, the knockdown efficiency of siRNAs was tested by real-time quantitative polymerase chain reaction (RT-qPCR) or Western blot assay.

## Cell counting kit-8 (CCK-8) method

Under the provided instructions, cell proliferation was analyzed using CCK-8 kits (Beyotime, Shanghai, China). After transfection, A375 or A2058 cells (at least 3 technical replicates) were loaded into the 96-well plates at a density of $5 \times 10^3$ cells/well and cultured for 0, 24, 48, and 72 h. At appointed time point, each well was added with 10 μL of CCK-8 reagent, followed by 2 h incubation in the incubator and measurement of absorbance at a wavelength of 450 nm with the help of a microplate reader (Bio-Rad, Hercules, CA, USA). This experiment was performed with 3 biological replicates.

## Colony formation

In colony formation assay, transfected A375 or A2058 cells were resuspended in a fresh medium. The cell suspension (2 mL) containing 1000 cells was loaded into 6-well plates and grown in a humidified air at 37˚C with 5% $CO_2$. After 2 weeks, colonies were treated with 4% paraformaldehyde for fixation and stained with 0.1% crystal violet. After that, colonies were imaged and counted under an inverted optical microscope (Olympus corporation, Tokyo, Japan). Eventually, the number of colonies containing at least 50 cells was counted manually.

## Transwell assay

For the migration assay, 48 h after transfection, A375 or A2058 cells were resuspended in a serum-free medium. Then, 200 μL of cell suspension containing $5 \times 10^4$ cells were loaded in apical chamber (BD Biosciences, San Jose, CA, USA), while 500 μL of complete medium was given into basolateral chamber. After one day incubation at 37˚C, non-migrative cells were wiped off using cotton swabs, and migrative cells were fixed with 4% paraformaldehyde and dyed with 0.1% crystal violet. For the invasion assay, the chamber was precoated with 150 μL Matrigel (2.5 mg/mL, BD Biosciences), and the remaining experimental procedures were as the same as the migration assay. At last, stained cells were observed under an inverted light microscope, and the number of migrative and invasive cells in 5 randomly selected visual fields was counted using ImageJ software.

## RNA immunoprecipitation (RIP) assay

Extracts from A375 or A2058 cells were used for RIP assay using the EZ-Magna RIP kits (17–701, Millipore, Billerica, MA, USA). A portion of supernatant was used as Input. The other portion of supernatant was incubated with 1 mg magnetic beads precoated with antibodies against IgG (ab172730, Abcam, Cambridge, MA, USA) or HuR (ab200342, Abcam) at 4˚C overnight. Next, the RNA compound was incubated with proteinase K (RIP washing buffer, 10% sodium dodecyl sulfate (SDS), 10 mg/mL proteinase K, Millipore) at 55˚C for 30 min to digest the remaining protein on the beads. Afterwards, immunoprecipitated RNA was purified and analyzed by RT-qPCR.

## RNA stability assay

To determine the half-life of ALKBH5 mRNA in A375 or A2058 cells, cells were added with actinomycin D (2 μg/mL, Sigma, St. Louis, MO, USA) and stayed for 0, 3, 6, and 12 h. At appointed time points, the total RNA was harvested and the remaining ALKBH5 mRNA was analyzed by RT-qPCR.

## Chromatin immunoprecipitation (ChIP) assay

ChIP assay was conducted applying the SimpleChIP® enzymatic chromatin IP kits (Cell Signaling Technology, Danvers, MA, USA). Chromatin was incubated with antibodies against H3K9me3 (ab8898, Abcam), or JMJD2C (PA5-23065, Thermo Fisher Scientific), or IgG (serving as negative control, ab172730, Abcam) for immunoprecipitation. Eventually, the compound of immunoprecipitated protein and DNA was extracted and used for RT-PCR to confirm the binding site. The extraction of DNA was conducted as follow: samples were rinsed in the lysis buffer twice, and were rinsed in 1 M lysis buffer (50 mM Tris, pH 7.4, 1 M NaCl, 1 mM EDTA, 0.1% SDS, 1% NP-40, 0.5% sodium deoxycholate) for 4 times, and were resuspended in lysis buffer, followed by 45 min treatment with proteinase K at 45˚C. Co-immunoprecipitated DNA was purified by QIAquick DNA purification columns (Qiagen, Germantown, MD, USA) and were eluted using 50 μL nuclease-free water. The primers of the ALKBH5 promoter were as follows: forward primer: 5'-TCTCCTTTAGGGGTCCTCGC-3', reverse primer: 3'GGAGTTTCCGGAAGTCGGTT-5'.

## Tumor xenograft assay

The protocol of animal experiments was ratified by the Ethics Committee of Zhujiang Hospital of Southern Medical University. All animal use, care, and operating procedures complied to the National Institutes of Health Guide for the Care and Use of Laboratory Animals [20]. The reporting of this study followed ARRIVE 2.0 guidelines [21]. Extensive measures were undertaken to reduce the animal number and suffering as much as possible. BALB/c nude female mice (6 weeks old, 16–18 g, Beijing Vital River Laboratory Animal Technology Co., Ltd, license No: SYXK(Beijing), 2017–0033) were housed under pathogen-free specific conditions at 20-22˚C and 55 ± 10% humidity with 12 h day/night cycles and were given free access to food and water. A375 cells ($2 \times 10^5$/well) were loaded in 6-well culture dishes and infected with 50 times multiplicity of infection (MOI) of MIR205HG-knockdown lentivirus (sh-MIR205HG) or its negative control (sh-NC) by following the manufacturer's instructions. After that, cells were given 3 μg/mL puromycin (Invitrogen, Carlsbad, CA, USA) for 2 weeks to screen stable knockout cells. Mice were numbered according to their weights and allocated to the control group and the treatment group based on the random number method, which was recorded by an experimenter. Each mouse with a total number of 12 was subcutaneously injected with stable infected A375 cells (about $4 \times 10^6$ cells, in 100 μL sterile phosphate buffer saline) in the right flank. The volume (V) of tumor was appraised by the tumor width (W) and length (L) and calculated according the formula: V = $(W^2 \times L)/2$. The tumor volume was measured every 7 days. The health and behavioral status of all animals was tested every two days, and nude mice were euthanized if the following conditions (humane endpoints) occurred: weight loss > 10% of body weight in nude mice, or animals suffering from tumor load, or tumors with a maximum diameter of more than 1.5 cm. No animals died during the experiment. At the 28th day after subcutaneous injection, mice were euthanatized using an intraperitoneal injection of pentobarbital sodium ($\geq$ 100 mg/kg). After resection of tumors, the tumor weight was recorded and immunohistochemistry and molecular assays were performed. To be specific, 6 tumors from each group were used for immunohistochemistry, which was analyzed

by two experts unknown of the experimental purpose using the double-blind method; the remaining 6 tumor from each group were used for RT-qPCR and Western blot assay.

## Immunohistochemistry

Tumors resected from the xenograft mouse model were fixed with 4% paraformaldehyde, embedded in paraffin, and cut into 5 μm sections. After de-paraffining and rehydration, sections were incubated with 3% $H_2O_2$ for 20 min to block the activity of endogenous peroxidase. Sections were blocked with 10% FBS at room temperature for 1 h and cultured with antibody against Ki67 (1:1000, ab15580, Abcam) at 4°C overnight, and then cultured with secondary antibody against IgG (1:1000, ab6721, Abcam) at room temperature for 30 min. Glass slide was treated with hematoxylin to counterstain nuclei. After that, sections were dehydrated, sealed with neutral glue, and observed under a microscope (Olympus CKX51) using the double-blind method.

## RT-qPCR

The total RNA was separated from cell lines and tumor tissues using the TRIzol reagent (Takara Bio, Inc., Tokyo, Japan). The PrimeScript RT kit (Takara Bio, Inc.) was applied to synthesis of the complementary DNA, and SYBR green (Takara Bio, Inc.) and ABI7500 real-time PCR system were used for qPCR. With GAPDH as the internal reference, the relative amount of gene expression was calculated according to the $2^{-\Delta\Delta Ct}$ method [22]. Primers of PCR are exhibited in Table 1.

## Western blot assay

The total protein was extracted using radioimmunoprecipitation assay buffer (Beyotime Institute of Biotechnology) containing 1% phenylmethanesulfonylfluoride and 1 mmol/L β-glycerophosphate sodium salt hydrate. The bicinchoninic acid protein quantification kit (Beyotime Institute of Biotechnology) was used to measure the protein concentration. After being separated by 10% SDS-polyacrylamide gel (Beyotime Institute of Biotechnology), protein samples were transferred onto nitrocellulose membranes (Millipore). The membranes were blocked with 5% skim milk at room temperature for 2 h, and incubated with primary antibodies against JMJD2C (PA5-23065, 1:10000, Thermo Fisher Scientific), HuR (ab200342, 1:1000, Abcam), and β-actin (ab8227, 1:1000, Abcam) at 4°C overnight and then incubated the secondary antibody (ab205718, 1:2000, Abcam) at room temperature for 1 h. Subsequently, the chemiluminescence signal were visualized using the enhanced chemiluminescence kit

**Table 1. Information of PCR primers.**

| Gene | Sequence (5'-3') |
|---|---|
| MIR205HG | F: TCCCTCTTGCTCACCCTTGA |
| | R: AGAGGGAGTAAAGGTAGCTGG |
| JMJD2C | F: TTGGGCGAAACCTCTCATCC |
| | R: TATCTGGCTTGTGGTACGGC |
| HuR | F: AGCTTGGGCTATGGCTTTGT |
| | R: GGCGAGCATACGACACCTTA |
| ALKBH5 | F: GCGCAAGTCATACGAGTCCT |
| | R: GCATCTTCACCTTTCGGGCA |
| GAPDH | F: GATGCTGGCGCTGAGTACG |
| | R: GCTAAGCAGTTGGTGGTGC |

(Millipore). Protein levels were appraised after exposure with X-ray film (Fuji Photo film Co., Ltd, China) and were normalized to β-actin, and analyzed using ImageJ software.

### Statistical analysis

Data statistical analysis and graphing were performed with application of SPSS21.0 statistical software (IBM Corp, Armonk, NY, USA) and GraphPad Prism 8.0 software (GraphPad Software Inc., San Diego, CA, USA). Data were normally distributed with equal variance as examined. Statistical differences between two panels were analyzed through the $t$ test, and statistical differences among multiple groups were analyzed through one-way or two-way analysis of variance (ANOVA), followed by Tukey's multiple comparison test. A value of $P < 0.05$ was indicative of statistical significance, and a value of $P < 0.01$ was indicative of highly statistical significance.

## Results

### MIR205HG downregulation inhibits proliferation of melanoma cells

It has reported that MIR205HG is upregulated in melanoma [13, 23]. The expression pattern of MIR205HG was determined in cultured cell lines and the results showed the upregulation of MIR205HG expression in melanoma cells ($P < 0.01$, Fig 1A). A375 cells with higher MIR205HG expression and A2058 cells with lower MIR205HG expression were selected to investigate the impact of MIR205HG on melanoma cell proliferation. The cells were transfected with siRNAs, resulting in knockdown of intercellular expression of MIR205HG ($P < 0.01$, Fig 1B), and as a result si-MIR205HG-1 and si-MIR205HG-3 with higher knockdown efficiency were selected for the subsequent experiments. Our results showed that after MIR205HG downregulation, cell proliferation was notably decreased ($P < 0.01$, Fig 1C and 1D).

### MIR205HG downregulation inhibits invasion and migration of melanoma cells

Subsequently, we evaluated the impact of MIR205HG on invasion and migration of melanoma cells by Transwell assays. Our results showed that MIR205HG downregulation inhibited invasion and migration of melanoma cells ($P < 0.01$, Fig 2A and 2B).

### MIR205HG binds to HuR and stabilizes JMJD2C expression

Long non-coding RNAs can bind to HuR to stabilize gene expression [14]. The interactions between MIR205HG and HuR, HuR and JMJD2C (KDM4C) were found on the StarBase database (http://starbase.sysu.edu.cn/index.php) [24]. JMJD2C upregulation has been documented in melanoma [16, 25]. In RIP assay, HuR recruited more MIR205HG or JMJD2C compared with IgG ($P < 0.01$, Fig 3A and 3B). The expression levels of JMJD2C were markedly increased in melanoma cells but were downregulated in response to MIR205HG downregulation ($P < 0.05$, Fig 3C and 3D). The mRNA stability of JMJD2C was reduced after MIR205HG downregulation ($P < 0.01$, Fig 3E). In addition, we upregulated HuR expression in melanoma cells ($P < 0.01$, Fig 3F and 3G), upon which the expression level and mRNA stability of JMJD2C were both elevated ($P < 0.05$, Fig 3C–3F). Above all, MIR205HG bound to HuR and stabilized JMJD2C expression.

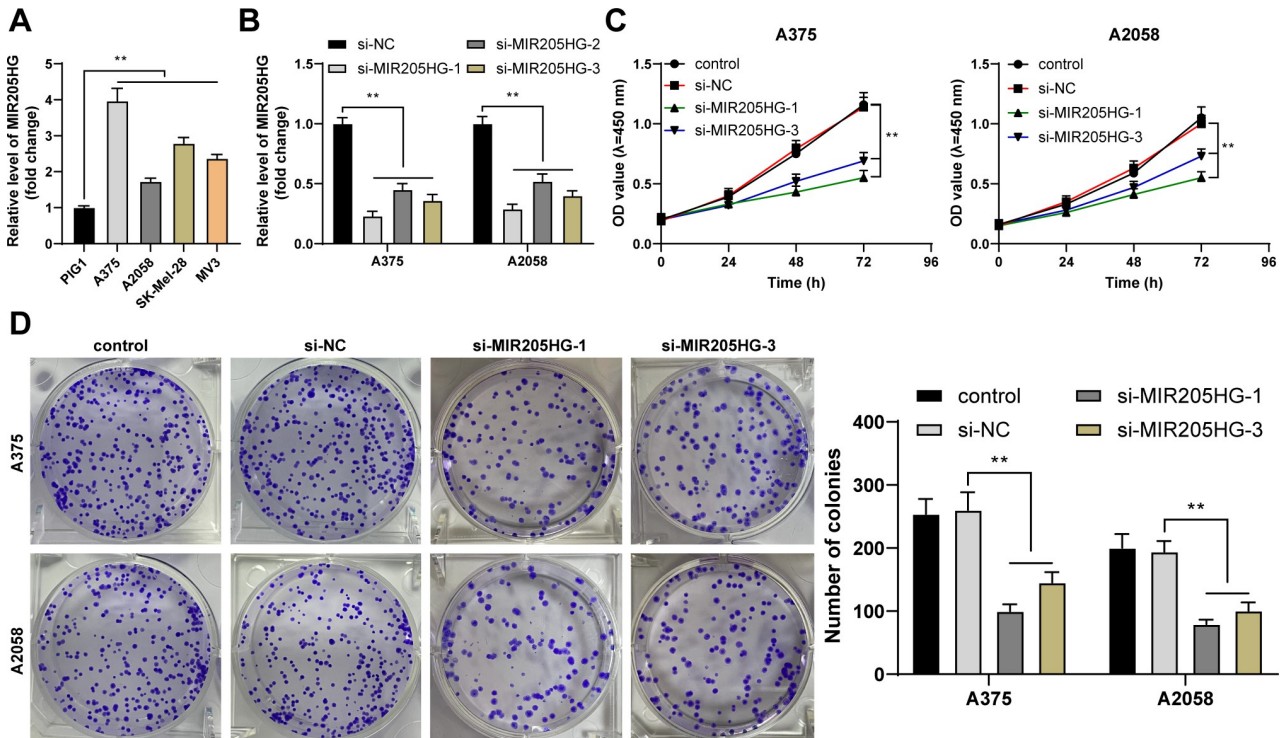

**Fig 1. MIR205HG downregulation inhibits proliferation of melanoma cells.** A: MIR205HG expression levels in cell lines were determined by RT-qPCR; MIR205HG siRNAs (si-MIR205HG-1, si-MIR205HG-2, and si-MIR205HG-3) were transfected into A375 and A2058 cells, with si-NC as negative control; B: Knockdown efficiency of MIR205HG siRNAs was determined by RT-qPCR; C, D: Cell proliferation was evaluated by CCK-8 assay (C) and colony formation assay (D). Cell experiments were performed in triplicate, and data were expressed as mean ± standard deviation. Multi-group comparisons in panel A were analyzed using one-way ANOVA, and multi-group comparisons in panels B-D were analyzed by two-way ANOVA, followed by Tukey's multiple comparison test. ** $P < 0.01$.

## JMJD2C reverses the inhibition of melanoma cell proliferation, invasion, and migration caused by silencing MIR205HG

To confirm the impact of JMJD2C on melanoma cell functions, JMJD2C expression was upregulated in A375 cells ($P < 0.01$, Fig 4A and 4B) and then A375 cells were further treated with si-MIR205HG-1 for rescue experiments. The results showed that JMJD2C overexpression effectively enhanced cell proliferation, invasion, and migration ($P < 0.01$, Fig 4C and 4F), but the potential for proliferation, invasion, and migration were lower in the JMJD2C overexpression group than the control and si-NC groups ($P < 0.05$, Figs 1C, 1D and 2A and 2B).

## JMJD2C reduces H3K9me3 enrichment in the ALKBH5 promoter region and promotes ALKBH5 transcription

It has been documented that JMJD2C decreases H3K9me3 level to improve ALKBH5 expression [17], and ALKBH5 is upregulated in melanoma [19]. In the ChIP assay, we observed that relative to IgG, JMJD2C and H3K9me3 were enriched in the ALKBH5 promoter region ($P < 0.01$, Fig 5A and 5B); relative to the si-NC group, JMJD2C enrichment in the ALKBH5 promoter region was decreased while H3K9me3 enrichment in the ALKBH5 promoter region was elevated in the si-MIR205HG group; relative to the si-MIR205HG + NC group, JMJD2C enrichment in the ALKBH5 promoter region was increased while H3K9me3 enrichment in the ALKBH5 promoter region was reduced in the si-MIR205HG + JMJD2C group ($P < 0.01$,

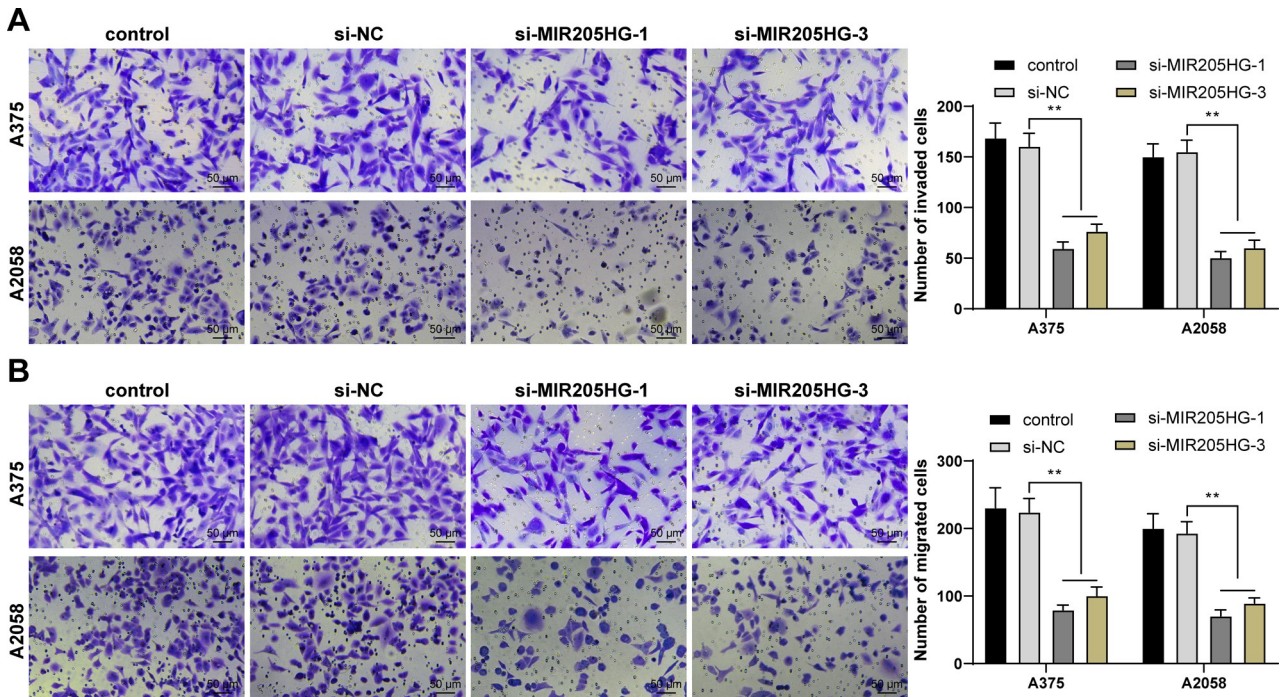

**Fig 2. MIR205HG downregulation inhibits invasion and migration of melanoma cells.** A, B: Invasion (A) and migration (B) of A375 and A2058 cells were evaluated by Transwell assays. Cell experiments were performed in triplicate, and data were expressed as mean ± standard deviation. Multigroup comparisons in panels A, B were analyzed using two-way ANOVA, followed by Tukey's multiple comparison test. ** $P < 0.01$.

Fig 5A and 5B). ALKBH5 expression levels were augmented in melanoma cells, and MIR205HG downregulation decreased the mRNA levels of ALKBH5 while JMJD2C overexpression promoted the mRNA levels of ALKBH5 ($P < 0.01$, Fig 5C and 5D). Above all, JMJD2C reduced H3K9me3 enrichment in the ALKBH5 promoter region and promoted ALKBH5 transcription.

## ALKBH5 overexpression reverses the inhibition of melanoma cell proliferation, invasion, and migration caused by silencing MIR205HG

To confirm the impact of ALKBH5 on melanoma cell functions, we upregulated ALKBH5 mRNA level ($P < 0.01$, Fig 6A) and cells with ALKBH5 upregulation were further treated with si-MIR205HG-1 for rescue experiments. The results showed that compared with MIR205HG downregulation alone, the combined treatment enhanced cell proliferation, invasion, and migration ($P < 0.01$, Fig 6B–6E), but the potential for proliferation, invasion, and migration were lower in the combined treatment group than the control and si-NC groups ($P < 0.05$, Figs 1C, 1D and 2A and 2B).

## Silencing MIR205HG inhibits tumor growth by downregulating JMJD2C/ALKBH5

At last, we strived to validate our mechanism *in vivo*. The xenograft mouse model was established through subcutaneous injection of A375 cells with MIR205HG knockdown. Our results revealed that MIR205HG downregulation inhibited tumor growth and reduced the positive rate of Ki67 in tumor tissues ($P < 0.01$, Fig 7A–7C). In addition, relative to the sh-NC group, the levels of MIR205HG, JMJD2C, and ALKBH5 in tumor tissues were diminished in the sh-

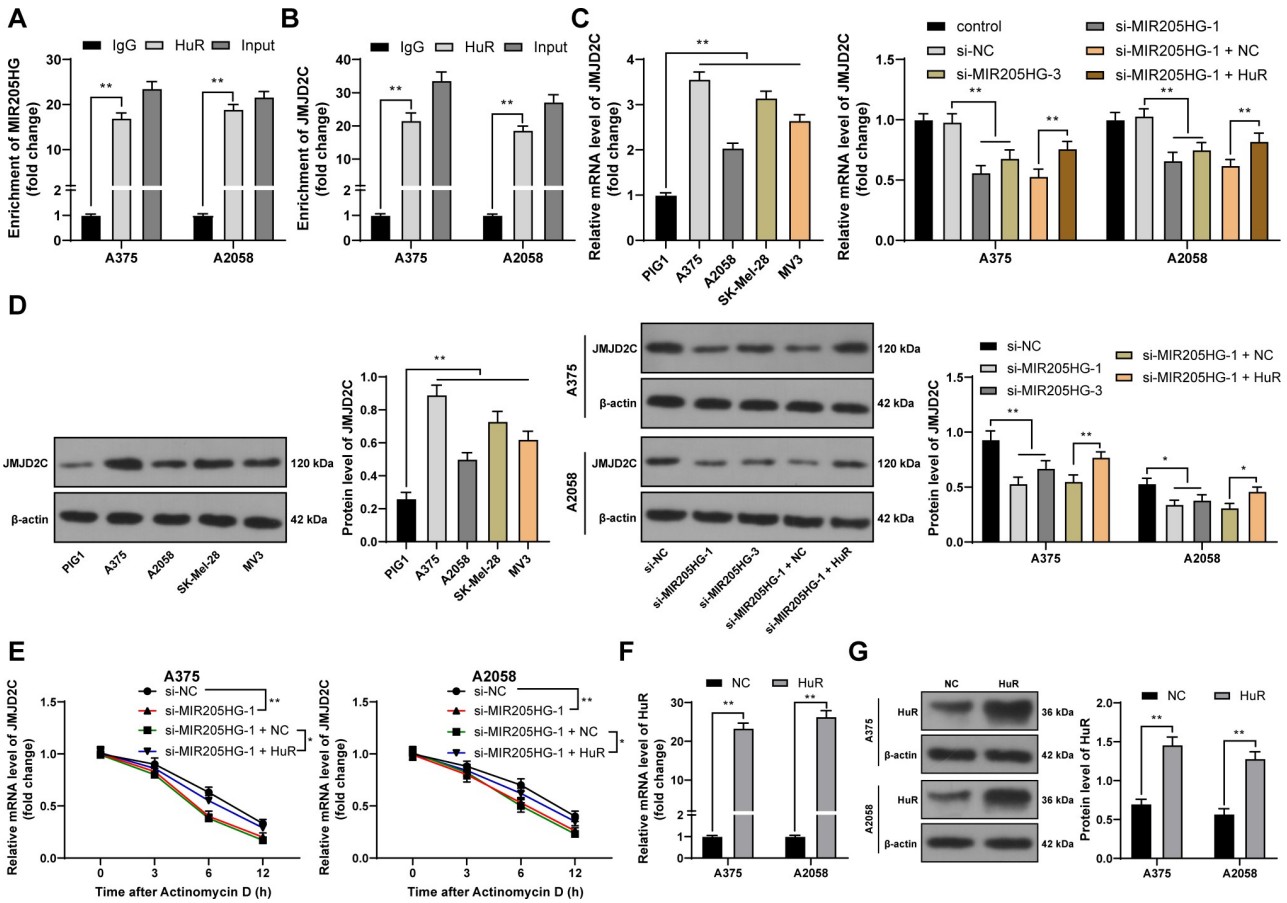

**Fig 3. MIR205HG binds to HuR and stabilizes JMJD2C expression.** A, B: Binding of MIR205HG to HuR (A) and HuR to JMJD2C (B) were analyzed by RIP; C, D: JMJD2C expression levels were determined by RT-qPCR (C) and Western blot assay (D); E: After actinomycin D treatment, JMJD2C mRNA levels were determined by RT-qPCR; Cells were transfected with pcDNA3.1-HuR (HuR), with pcDNA3.1 empty vector (NC) as negative control; F-G: HuR expression levels were determined by RT-qPCR (F) and Western blot assay (G). Cell experiments were performed in triplicate, and data were expressed as mean ± standard deviation. Multi-group comparisons in panels A-G were analyzed using two-way ANOVA, and multi-group comparisons in panels C, D were analyzed using one-way ANOVA, followed by Tukey's multiple comparison test. * $P < 0.05$, ** $P < 0.01$.

MIR205HG group ($P < 0.01$, Fig 7D–7G). Above all, our findings suggested that silencing MIR205HG inhibited tumor growth by downregulating JMJD2C/ALKBH5.

## Discussion

Melanoma is a highly malignant skin cancer, with its incidence increasing rapidly worldwide [26]. Data are accumulating that lncRNAs are crucial players in melanoma growth, metastasis, and drug-resistance [27–29]. In addition, lncRNAs exert clinical implication regarding the prediction of the prognosis and outcome of immunotherapy in melanoma patients [30]. Herein, our findings suggested that 1) MIR205HG, JMJD2C, and ALKBH5 were highly expressed in melanoma; 2) MIR205HG may stabilize JMJD2C expression by binding to HuR; 3) JMJD2C may reduce the occupation of H3K9me3 in the ALKBH5 promoter to promote ALKBH5 transcription, thus boosting proliferation, invasion, and migration of melanoma cells (Fig 8).

Liu et al have unveiled that MIR205HG plays a significant performance in prognosis of melanoma [31]. Guo et al also have indicated that MIR205HG is upregulated in melanoma cells [13]. In this study, MIR205HG expression was found to be higher in melanoma cell lines and

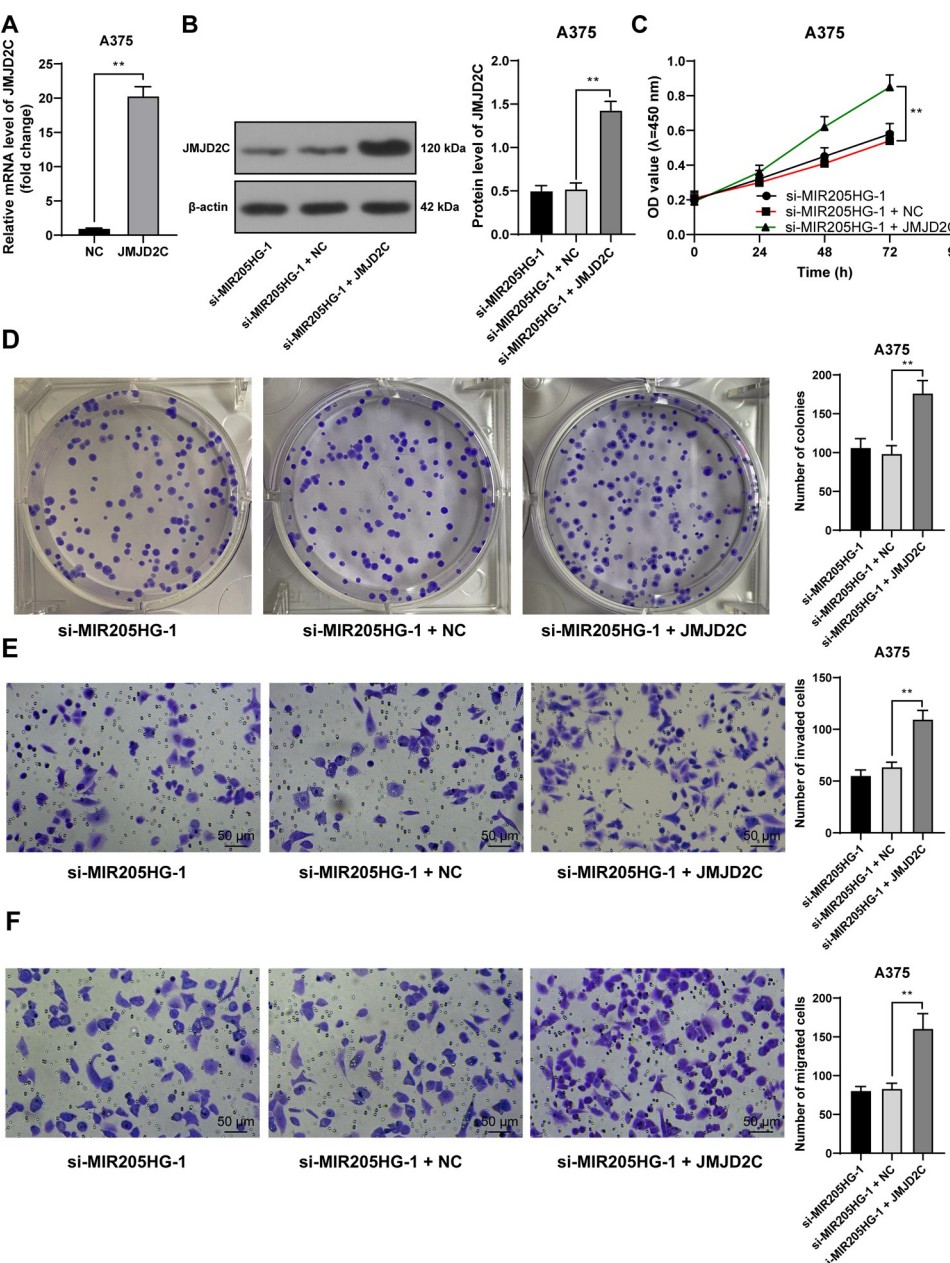

**Fig 4. JMJD2C reverses the inhibition of melanoma cell proliferation, invasion, and migration caused by silencing MIR205HG.** Cells were transfected with pcDNA3.1-JMJD2C (JMJD2C), with pcDNA3.1 empty vector (NC) as negative control. A, B: JMJD2C expression levels were determined by RT-qPCR (A) and Western blot assay (B); C, D: Cell proliferation was evaluated by CCK-8 assay (C) and colony formation assay (D); E, F: Cell invasion (E) and migration (F) were evaluated by Transwell assays. Cell experiments were performed in triplicate, and data were expressed as mean ± standard deviation. Pairwise comparisons in panel A were analyzed by the *t* test, multi-group comparisons in panels B, and D-F were analyzed using one-way ANOVA, and multi-group comparisons in panel C were analyzed using two-way ANOVA, followed by Tukey's multiple comparison test. ** $P < 0.01$.

MIR205HG downregulation inhibited proliferation, invasion, and migration of melanoma cells. In agreement with our results, MIR205HG has been demonstrated to accelerate the proliferation of melanoma cells by inducing the canonical oncogenic Wnt/β-catenin signaling pathway [32] and support melanoma tumor growth by mediating the miR-299-3p/vascular

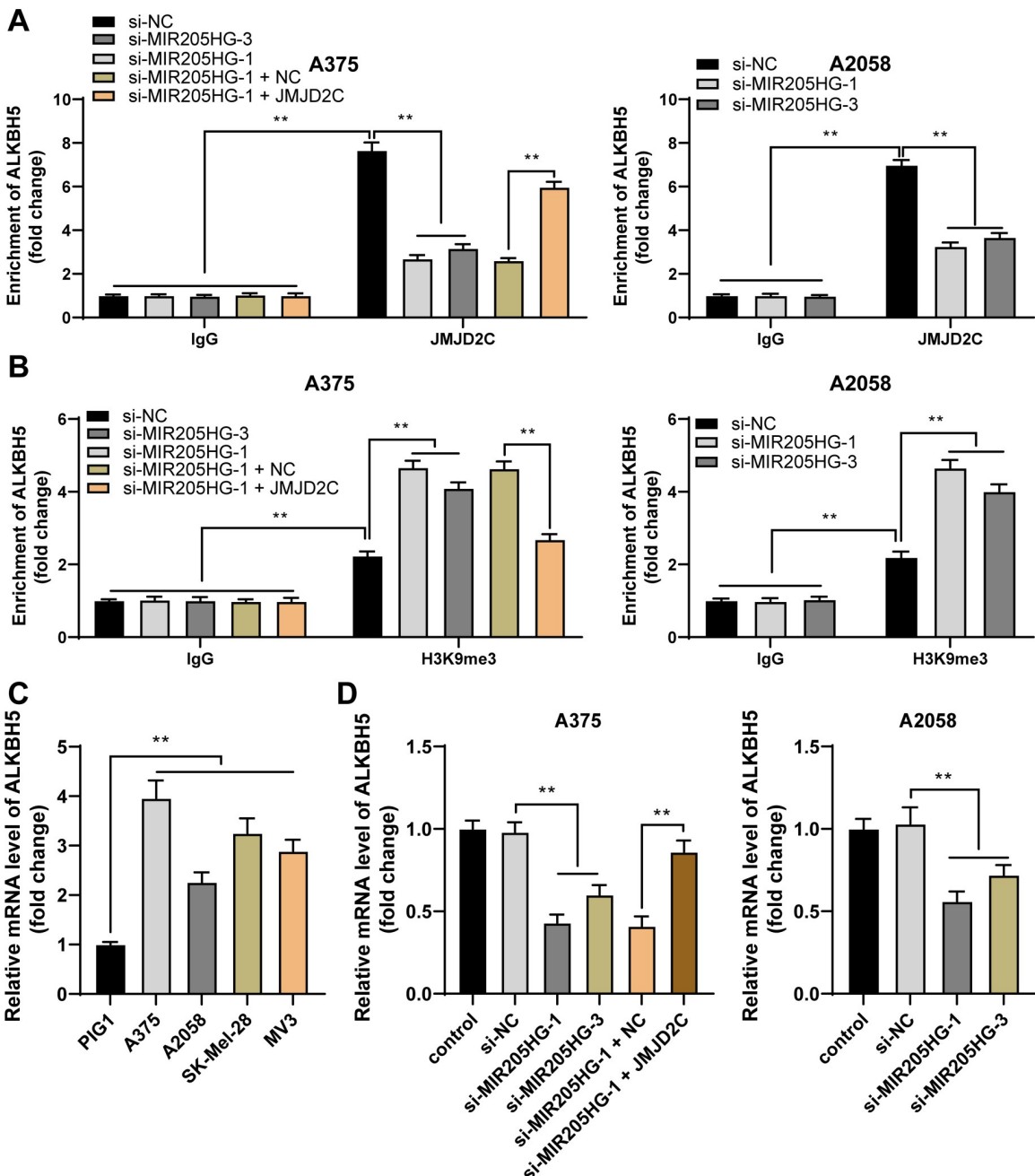

**Fig 5. JMJD2C reduces H3K9me3 enrichment in the ALKBH5 promoter region and promotes ALKBH5 transcription.** A, B: JMJD2C (A) or H3K9me3 (B) enrichment in the ALKBH5 promoter region was analyzed by the ChIP assay. C, D: ALKBH5 mRNA levels were quantified by RT-qPCR. Cell experiments were performed in triplicate, and data were expressed as mean ± standard deviation. Multi-group comparisons in panels A, B were analyzed using two-way ANOVA, and multi-group comparisons in panels C, D were analyzed using one-way ANOVA, followed by Tukey's multiple comparison test. * $P < 0.05$, ** $P < 0.01$.

endothelial growth factor A axis [13]. Likewise, MIR205HG also exerts oncogenic effects in cervical cancer, esophageal squamous cell carcinoma, and osteosarcoma [11, 12, 33]. Moreover, lncRNAs can stabilize gene expression by promoting the recognition of HuR to its targets [14]. JMJD2C correlates with the epigenetic regulation in various cancer types [34–36]. The StarBase databases and RIP assay revealed the binding of MIR205HG to HuR and HuR to

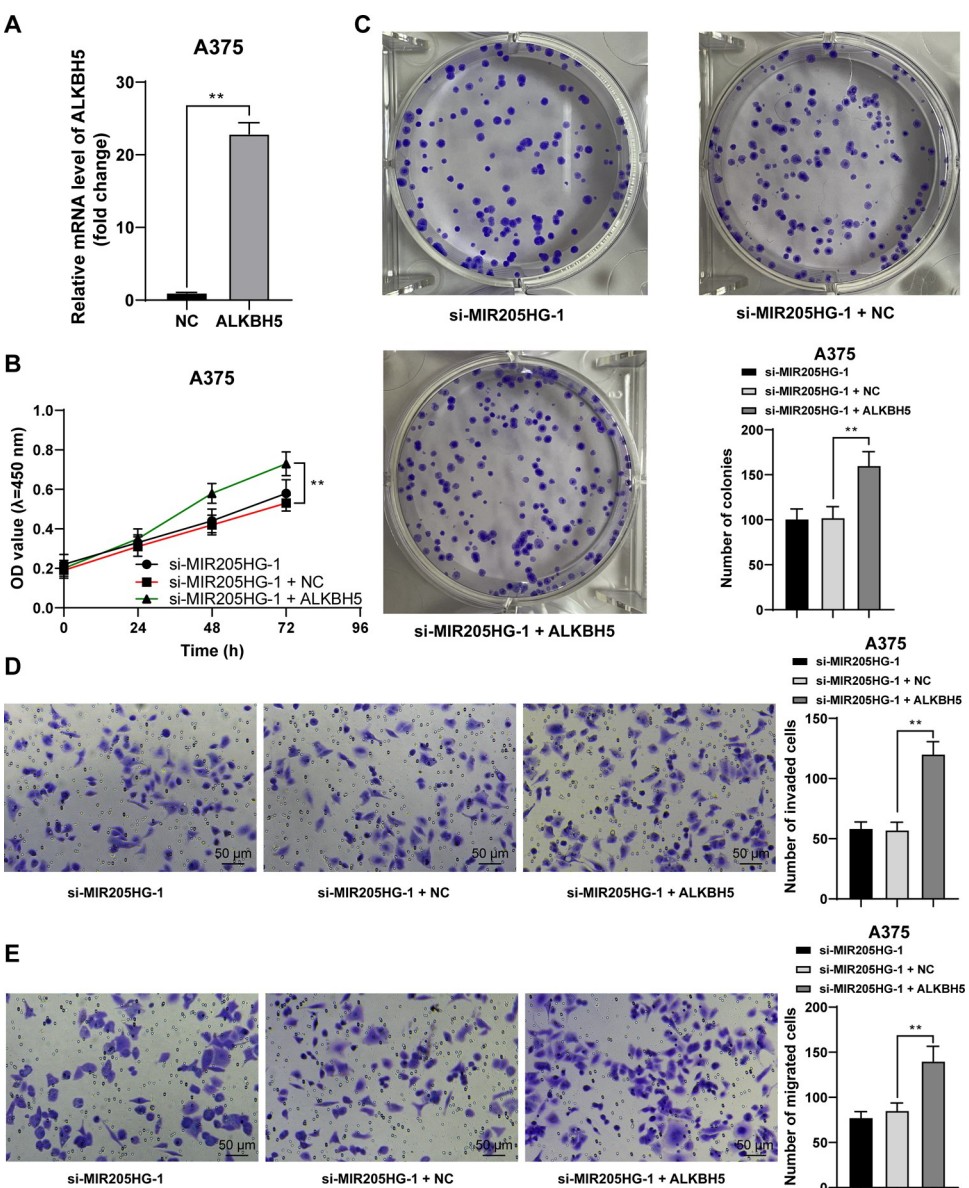

**Fig 6. ALKBH5 overexpression reverses the inhibition of melanoma cell proliferation, invasion, and migration caused by silencing MIR205HG.** Cells were transfected with pcDNA3.1-ALKBH5 (ALKBH5), with pcDNA3.1 empty vector (NC) as negative control. A: ALKBH5 expression levels were determined by RT-qPCR; B, C: Cell proliferation was evaluated by CCK-8 assay (B) and colony formation assay (C); D, E: Cell invasion (D) and migration (E) were evaluated by Transwell assays. Cell experiments were performed in triplicate, and data were expressed as mean ± standard deviation. Pairwise comparisons in panel A were analyzed by the *t* test, multi-group comparisons in panels C-E were analyzed using one-way ANOVA, and multi-group comparisons in panel B were analyzed using two-way ANOVA, followed by Tukey's multiple comparison test. ** $P < 0.01$.

JMJD2C. JMJD2C was found to be higher in melanoma cell lines, and MIR205HG downregulation reduced the expressive level and mRNA stability of JMJD2C, while HuR upregulation played an opposite role, indicating that MIR205HG bound to HuR and stabilized expression. Additionally, JMJD2C overexpression aggravated the proliferation, invasion, and migration of melanoma cells. Consistently, enforced JMJD2C expression has been known to promote melanomagenesis and drug-resistance of melanoma cells [16, 25]. Collectively, our findings initially

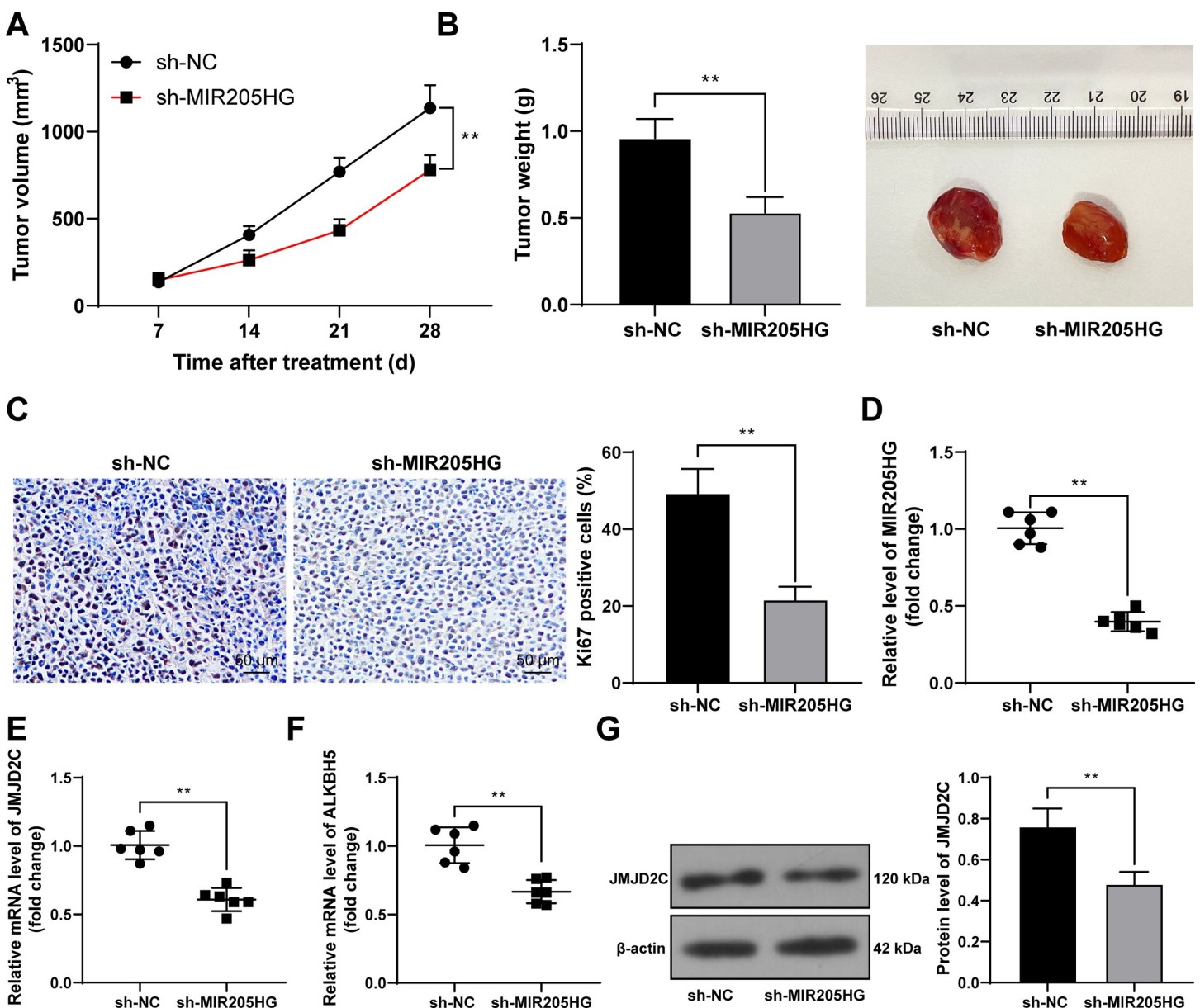

**Fig 7. Silencing MIR205HG inhibits tumor growth via downregulating JMJD2C/ALKBH5.** The xenograft mouse model was established using A375 cells with MIR205HG knockdown. A, B: Tumor volume and weight (representative pictures of tumors); C: Positive rate of Ki67 in tumor tissues was tested by immunohistochemistry; D-F: Levels of MIR205HG, JMJD2C, and ALKBH5 in tumor tissues were determined by RT-qPCR; G: JMJD2C protein levels in tumor tissues were determined by Western blot assay. n = 6, cell experiments were performed in triplicate, and data in panels A-C and G were expressed as mean ± standard deviation. Pairwise comparisons in panels B-G were analyzed by the *t* test, and multi-group comparisons in panel A were analyzed using two-way ANOVA, followed by Tukey's multiple comparison test. ** *P* < 0.01.

demonstrated that MIR205HG played an oncogenic role in melanoma through HuR-mediated stabilization of JMJD2C.

JMJD2C, also known as KDM4C, is a H3K9me3-specific histone demethylase [37]. JMJD2C has been documented to erase H3K9me3 to enhance ALKBH5 expression [17]. ALKBH5 regulates the progression of cancers in a cell type-dependent manner. For instance, in gastric, colon, and breast cancers [38–40], ALKBH5 promotes tumorigenesis, whereas in esophageal, hepatocellular carcinoma, osteosarcoma, ALKBH5 reduces cancer malignancy

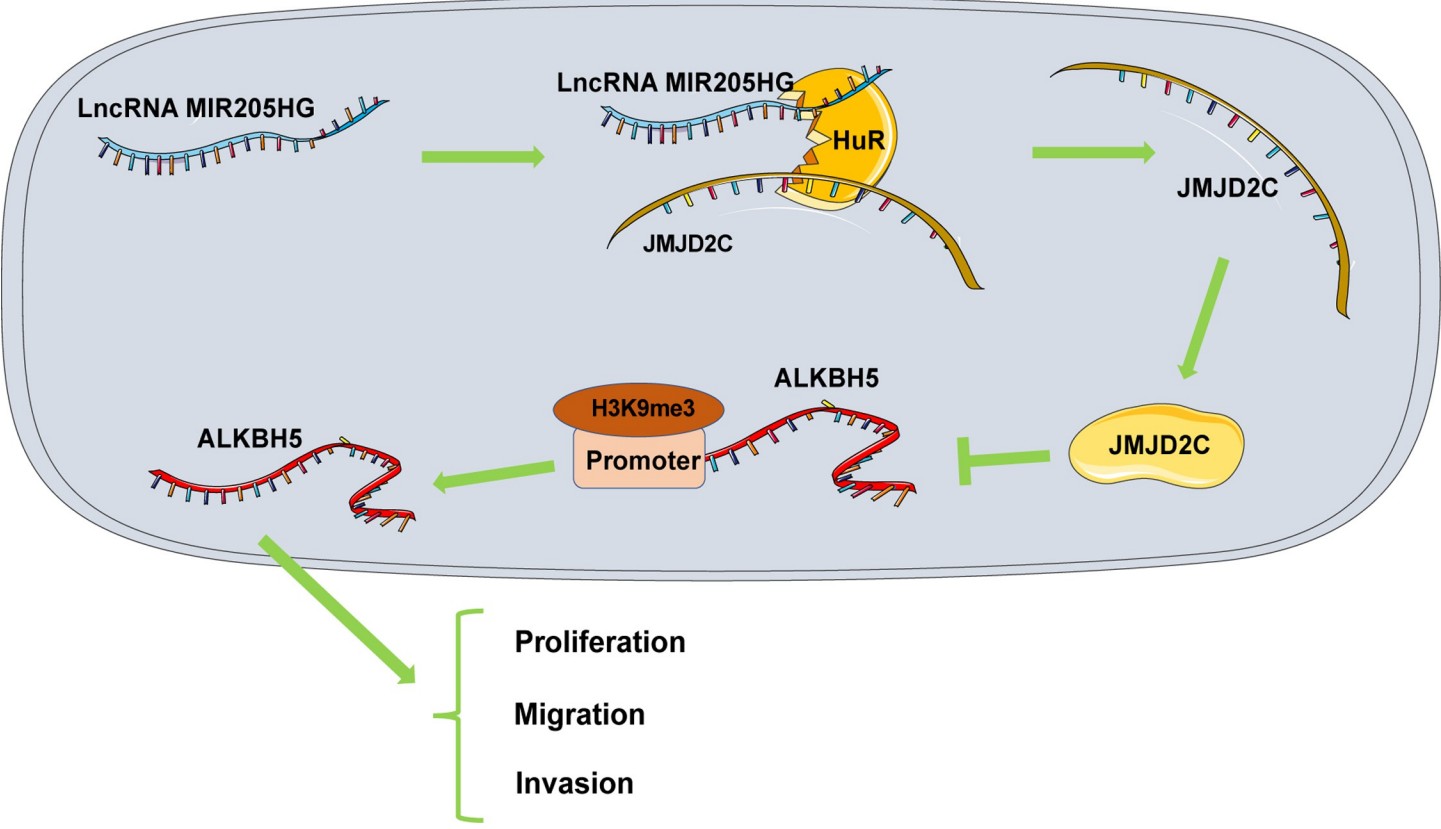

**Fig 8. Molecular mechanism of MIR205HG in proliferation, invasion, and migration of melanoma cells.** MIR205HG upregulation may stabilize JMJD2C expression by binding to HuR and reduce H3K9me3 enrichment in the ALKBH5 promoter region to promote ALKBH5 transcription, thus facilitating proliferation, invasion, and migration of melanoma cells.

[41–43]. According to our results, high ALKBH5 expression was found in melanoma cell lines. In ChIP assay, relative to IgG, JMJD2C and H3K9me3 were enriched in the ALKBH5 promoter region, but relative to the si-MIR205HG + NC group, JMJD2C enrichment in the ALKBH5 promoter region was increased while H3K9me3 enrichment in the ALKBH5 promoter region was reduced in the si-MIR205HG + JMJD2C group, suggesting that JMJD2C may reduce H3K9me3 enrichment in the ALKBH5 promoter region to promote ALKBH5 transcription. The rescue experiments based on pcDNA3.1-ALKBH5 and si-MIR205HG-1 disclosed that ALKBH5 overexpression reversed the inhibitory role of silencing MIR205HG-1 in proliferation, invasion, and migration of melanoma cells. In favor of our results, previous investigation has uncovered that ALKBH5 expedites the growth and metastasis of melanoma cells via m$^6$A demethylation of forkhead box M1 [19], and confers anti-tumor immunity for melanoma cells [44]. Thereafter, we established the xenograft mouse model and found that MIR205HG downregulation suppressed tumor growth and reduced the positive rate of Ki67, together with reductions in JMJD2C and ALKBH5 levels in tumors. Overall, our findings initially disclosed that MIR205HG downregulation retarded melanoma cell proliferation, invasion, migration, and *in vivo* tumor growth by repressing the JMJD2C/ALKBH5 axis.

However, our study has multiple limitations. The downstream of MIR205HG has many RNA-binding proteins (RBPs), therefore whether MIR205HG can regulate downstream gene expression via other RBPs apart from HuR needs further investigation. We did not determine the protein levels of ALKBH5 in melanoma cell lines. In the animal assay, we did not evaluate

the role of MIR205HG in invasion and migration of melanoma cells *in vivo*. In the future, we will explore the more well-round role and mechanism of MIR205HG in melanoma cell functions, collect clinical samples to validate our mechanism, and evaluate the role of ALKBH5 protein level in melanoma, so as to provide more theoretical references for melanoma treatment.

## Conclusion

In essence, our finding demonstrated that MIR205HG may stabilize JMJD2C expression via binding to HuR and reduce the occupation of H3K9me3 in the ALKBH5 promoter to promote ALKBH5 transcription, thus promoting proliferation, invasion, and migration of melanoma cells, and MIR205HG downregulation suppresses tumor growth *in vivo* via repressing the JMJD2C/ALKBH5 axis. Thus, MIR205HG, JMJD2C, and ALKBH5 may be candidate therapeutic targets for melanoma treatment.

## Supporting information

**S1 Raw images.**
(PDF)

**S2 Raw images.**
(PDF)

**S1 Data.**
(XLSX)

**S2 Data.**
(XLSX)

## Author Contributions

**Conceptualization:** Yujing Liu.

**Data curation:** Yujing Liu.

**Formal analysis:** Yujing Liu.

**Funding acquisition:** Lu Yan.

**Investigation:** Shanshan Wei.

**Methodology:** Suihai Wang.

**Writing – original draft:** Suihai Wang, Shanshan Wei, Xianwen Qiu, Yijie Mei.

**Writing – review & editing:** Lu Yan.

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
