## [Decision Letter · Decision Letter 0]

19 Jun 2023

PONE-D-23-08794The promotive role of lncRNA MIR205HG in proliferation, invasion, and migration of melanoma cells via the JMJD2C/ALKBH5 axisPLOS ONE

Dear Dr. Lu Yan

Thank you for submitting your manuscript to PLOS ONE. After careful consideration, we feel that it has merit but does not fully meet PLOS ONE’s publication criteria as it currently stands. Therefore, we invite you to submit a revised version of the manuscript that addresses the points raised during the review process.

ACADEMIC EDITOR: The comments from peer reviewers have been forwarded to you as listed below.

We look forward to receiving your revised manuscript.

Kind regards,

Xiangning Zhang, M.D., Ph.D.

Academic Editor

PLOS ONE

Journal Requirements:

“This study was supported in part by grants from National Natural Science Foundation of China (Grant number 82003370) and Natural Science Foundation of Guangdong Province (Grant number 2019A1515012225).”

Additional Editor Comments (if provided):

Re: The promotive role of lncRNA MIR205HG in proliferation, invasion, and migration of melanoma cells via the JMJD2C/ALKBH5 axis by Lu Yan et al.

Dear Dr Lu Yan,

Your above-referenced maunscript submitted to PlosOne has been received and reviewed by two external peer reviewers. Now that required number of reviews has been received, the referees' opinions are forwarded to you. Please revise your paper according to the comments and return its revised version to the editor at due time for further reviews.

Best,

Xiangning Zhang, M.D., Ph.D.

Academic Editor.

Reviewers' comments:

Reviewer's Responses to Questions

**Comments to the Author**

1. Is the manuscript technically sound, and do the data support the conclusions?

Reviewer #1: Yes

Reviewer #2: Yes

2. Has the statistical analysis been performed appropriately and rigorously? 

Reviewer #1: Yes

Reviewer #2: Yes

3. Have the authors made all data underlying the findings in their manuscript fully available?

Reviewer #1: Yes

Reviewer #2: No

4. Is the manuscript presented in an intelligible fashion and written in standard English?

Reviewer #1: Yes

Reviewer #2: Yes

5. Review Comments to the Author

Reviewer #1: In this manuscript, Liu et al present compelling evidence in support of their mechanism of MIR205HG effect on melanoma growth and progression. They show that this lncRNA acts via binding to JMJD2C and HuR, leading to upregulation of ALKBH5 and induction of pro-cancer phenotype. Overall, the manuscript is well-organized, and the data and conclusions presented with logical flow. Descriptions of statistical analyses are excellent. The manuscript does require some revision for clarity, both in terms of English word use and descriptions of the studies. The data presented in Figures 4 and 6 may benefit from inclusion of additional control data if available. Otherwise, specific minor queries and suggestions are given below:

Abstract:

-Line 26 – Change “(JMJD2C)/ALKB…” to “(JMJD2C) and ALKB…” As written, it implies these two proteins are the same.

-Line 31 – Please define HuR when first introduced.

Introduction:

-Line 46 – “Trendy” should be changed to “trending”.

-Line 57 – Reference #7 does not appear to cover all of the processes in this sentence; consider adding additional references.

Materials and Methods:

-Line 94 – Please describe transfection reagent and protocol used, including size of vessel and cell number.

-Line 109 – How long after transfection were colony assays seeded?

-Colony/transwell assays – Were cells counted manually or by software? How many visual fields?

-Line 117 – Please describe volume and composition of Matrigel mix added.

-Line 126 – Was RNA purification part of the kit? If not, how was this done?

-Line 134 – “Chromatins were” should read “Chromatin was.”

-Line 136-138 – How was DNA extracted? Also, the authors state it was extracted “to determine the number of specific proteins”. It would be more accurate to say the objective was to identify binding sites.

-Line 139 – Please verify the ALKBH5 promoter forward primer sequence. BLAST only identifies a partial match on Chromosome 17.

-Line 148 – “were free to food and water” should read “were given free access to food and water.” Also, please provide cell number and lentiviral dosage.

-Line 156 – “formular” should read “formula.”

-Line 171 – Please provide antibody dilutions for IHC.

-Line 193 – How were ECL signals captured? Film or imager?

Results:

-On line graphs such as Figure 1C, different conditions are difficult to tell apart. Authors may wish to enlarge the symbols and/or use color.

-Figure 3A – Please provide more info on the meaning of this panel, or else remove it. It may be sufficient to state in the text that these binding interactions are found in StarBase.

-Figure 3D – The color in the legend seems off. Please verify labeling of conditions or consider use of color.

-In Figures 4 and 6, it would be nice to compare the rescue by JMJD2C or ALKBH5 addback to control or si-NC treated cells. If these data are available, the authors may wish to include them. If not, they may wish to refer in the text to the degree of rescue and refer back to other figures where controls are present.

-Figure 5A and 5B – The order of the legend and the bars on the plot differ. Given how hard the greys are to tell apart, please remedy this.

-Why were JMJD2C overexpression studies not done in A2058 cells and added to Fig 5D?

-Line 238 – The authors state that H3K9me3 is enriched at the ALKBH5 promoter, but their argument is that this mark is being removed. Is this simply residual and it was previously higher? This is confusing.

-Lines 253 and 507– The phrase “stable low expression” is confusing. Consider “MIR205HG knockdown”.

-Figure 7D going over two lines is awkward. Authors may wish to consider splitting it into three individual panels D-F and relabeling 5E as 5G.

Discussion:

-Lines 264 and 316 – Please break up these sentences into smaller parts. Excellent use of Fig. 8 however.

-Line 293 – Similar to the note above re Line 238, the statement that H3K9me3 is enriched is confusing and seemingly contradicted in the next sentence. Please clarify.

-Line 307 – Why was ALKBH5 protein level not analyzed? If possible/practical, including these data would add to the evidence already described.

Figure Legends:

-Line 459 – What is shown is not transfection efficiency but knockdown efficiency.

-Are CCK-8 assays done with both biological and technical replicates?

-Figure 4 and 6 titles – “by silencing MIR205HG” at the end of the sentence is misleading. Please revise.

-Line 513 – the name of the statistical analysis used was omitted. Please revise.

Reviewer #2: The study is generally interesting and performed in a logical thought. However, it could be further improved from the following aspects.

1. In many places such as Line 243-244 (JMJD2C reduced H3K9me3 enrichment in the ALKBH5 promoter 244 region and promoted ALKBH5 transcription.), the tone should be softer since the conclusion was just based on the correlation analysis and literature. The direct effect and relationship among JMJD2C, H3K9me3 and ALKBH5 were not proved by the experiments.

2. The language needs to be polished by some native speaker.

3. More original data such as CHIP, RIP should be provided as supplementary materials in the manuscript.

6. PLOS authors have the option to publish the peer review history of their article (what does this mean?). If published, this will include your full peer review and any attached files.

Reviewer #1: **Yes: **Cai M. Roberts

Reviewer #2: No

---

## [Author Response · Author response to Decision Letter 0]

10 Aug 2023

Dear Dr. Xiangning Zhang and reviewers:

Thank you very much for your precious comments and advice. Those comments are all valuable and very helpful for improving our paper. We have studied comments carefully and have made correction which we hope meet with approval. Revised portion are marked in red in the paper. The main corrections in the paper and the responses to the reviewers’ comments are as flowing:

Responses to the reviewers’ comments:

Comments to the Author

1. Is the manuscript technically sound, and do the data support the conclusions?

Reviewer #1: Yes

Reviewer #2: Yes

2. Has the statistical analysis been performed appropriately and rigorously?

Reviewer #1: Yes

Reviewer #2: Yes

3. Have the authors made all data underlying the findings in their manuscript fully available?

Reviewer #1: Yes

Reviewer #2: No

4. Is the manuscript presented in an intelligible fashion and written in standard English?

Reviewer #1: Yes

Reviewer #2: Yes

5. Review Comments to the Author

Response: We very appreciate your careful reading and efforts to our manuscript. We have made corresponding revisions in the manuscript according to reviewers’ comments and made responses to their comments. Please review our manuscript again. Thanks.

Reviewer #1: In this manuscript, Liu et al present compelling evidence in support of their mechanism of MIR205HG effect on melanoma growth and progression. They show that this lncRNA acts via binding to JMJD2C and HuR, leading to upregulation of ALKBH5 and induction of pro-cancer phenotype. Overall, the manuscript is well-organized, and the data and conclusions presented with logical flow. Descriptions of statistical analyses are excellent. The manuscript does require some revision for clarity, both in terms of English word use and descriptions of the studies. The data presented in Figures 4 and 6 may benefit from inclusion of additional control data if available. Otherwise, specific minor queries and suggestions are given below:

Response: Thanks for your reading that helps improve the quality of the manuscript. We have made revisions in the manuscript according to your question and suggestion. As you mentioned that the data presented in Figures 4 and 6 may benefit from inclusion of additional control data if available, we made the following explanation. We have presented the impact of si-NC (the control of si-MIR205HG-1) on cells in figure 2. To avoid repetition of experimental results, we did not present relevant data of si-NC in figures 4 and 6. Instead, we presented data of control NC of JMJD2C overexpression and ALKBH5 overexpression in figures 4 and 6. Next, we would respond to your question one by one. Please review our manuscript again. Thanks again for your help with our manuscript.

Abstract:

-Line 26 – Change “(JMJD2C)/ALKB…” to “(JMJD2C) and ALKB…” As written, it implies these two proteins are the same.

Response: We have revised it according to your suggestion.

-Line 31 – Please define HuR when first introduced.

Response: We have provided the expansion of HuR when first introduced. Thanks.

Introduction:

-Line 46 – “Trendy” should be changed to “trending”.

Response: We have revised this word according to your suggestion.

-Line 57 – Reference #7 does not appear to cover all of the processes in this sentence; consider adding additional references.

Response: Thanks for your careful reading. We have replaced the reference here.

Materials and Methods:

-Line 94 – Please describe transfection reagent and protocol used, including size of vessel and cell number.

Response: Thanks for your question. We have added relevant information in the manuscript. Please review it again. Thanks.

-Line 109 – How long after transfection were colony assays seeded?

-Colony/transwell assays – Were cells counted manually or by software? How many visual fields?

Response: Thanks for your review. We performed colony assay 48 h after celltransfection. We have made explanation in the methods section. Colony formation was analyzed by manual counting, while cells in 5 randomly selected visual fields in Transwell assay were counted by ImageJ software. Relevant information have been supplemented in the manuscript.

-Line 117 – Please describe volume and composition of Matrigel mix added.

Response: According to your suggestion. We have supplemented the volume of Matrigel. In addition, according to the Matrigel purchase website, we knew that the Matrigel is derived from the basement membrane matrix of the Engelbreth-Holm-Swarm (EHS) mouse sarcoma, which contains about 60% laminin, 30% collagen IV and 8% nidogen, and basement membrane glycans, TGF-β, epidermal growth factor, insulin-like growth factor, tissue plasminogen and other growth factors. 

-Line 126 – Was RNA purification part of the kit? If not, how was this done?

Response: Thanks for your review. The principle of RIP assay kit is that after cell lysis, co-immunoprecipitation was conducted using specific antibody of RNA binding protein, followed by RNA separation and purification and analysis and characterization of RNA with help of RT-PCR. The assay kit we used contains RNA purification. Thanks.

-Line 134 – “Chromatins were” should read “Chromatin was.”

Response: We have revised this sentence according to your suggestion. Thanks for your reading.

-Line 136-138 – How was DNA extracted? Also, the authors state it was extracted “to determine the number of specific proteins”. It would be more accurate to say the objective was to identify binding sites.

Response: Thanks for your suggestion. We have added the method how to extract DNA and revised relevant descriptions. Please review it again. Thanks.

-Line 139 – Please verify the ALKBH5 promoter forward primer sequence. BLAST only identifies a partial match on Chromosome 17.

Response: We are very sorry that we made confusions when providing primers. We have made corrections of primer sequences.

-Line 148 – “were free to food and water” should read “were given free access to food and water.” Also, please provide cell number and lentiviral dosage.

Response: Thanks for your guidance. We have revised the linguistic mistake and stated the cell number and lentiviral dosage in the manuscript. Please review it again. Thanks.

-Line 156 – “formular” should read “formula.”

Response: We have revised the linguistic mistake. Thanks.

-Line 171 – Please provide antibody dilutions for IHC.

Response: According to your suggestion. We have stated the antibody dilutions in the manuscript. Thanks.

-Line 193 – How were ECL signals captured? Film or imager?

Response: We adopted X-ray film to capture ECL signals. We have stated it in the manuscript. Thanks.

Results:

-On line graphs such as Figure 1C, different conditions are difficult to tell apart. Authors may wish to enlarge the symbols and/or use color.

Response: According to your suggestion, we have enlarged the symbols and presented them with color in the line graph. Please read then again. Thanks.

-Figure 3A – Please provide more info on the meaning of this panel, or else remove it. It may be sufficient to state in the text that these binding interactions are found in StarBase.

Response: Thanks for your guidance. We have deleted the original figure 3A and written that the binding interaction was found in StarBase.

-Figure 3D – The color in the legend seems off. Please verify labeling of conditions or consider use of color.

Response: We have adjusted the color of histograms in figure 3 according to your suggestion. Thanks.

-In Figures 4 and 6, it would be nice to compare the rescue by JMJD2C or ALKBH5 addback to control or si-NC treated cells. If these data are available, the authors may wish to include them. If not, they may wish to refer in the text to the degree of rescue and refer back to other figures where controls are present.

Response: According to your question, we made the following explanation. We presented relevant data of control and si-NC treated cells in figure 2. To avoid repetition of experimental results, we did not present the relevant data of control and si-NC groups in figures 4 and 6. However, according to your suggestion, we stated the degree of rescue and indicated control in the manuscript. Please review it again. Thanks.

-Figure 5A and 5B – The order of the legend and the bars on the plot differ. Given how hard the greys are to tell apart, please remedy this.

Response: Given that you have mentioned many times that the color of the bar graphs is difficult to distinguish, we have made color changes in figures. Please read our figures again. Thanks.

-Why were JMJD2C overexpression studies not done in A2058 cells and added to Fig 5D?

Response: Thanks for your reading. We verified the effects of MIR205HG on A375 and A2058 cells in figures 1 and 2 and analyzed MIR205HG-mediated regulation of JMJD2C in two cell lines in figure 3. To further validate the effects of JMJD2C on MIR205HG-regulated cell functions, we selected A375 cells that have relatively high expression of both MIR205HG and JMJD2C. Meanwhile, due to limitation of experimental fund, we did not verify the mechanism in A2058 cells. Therefore, relevant results cannot be presented in figure 5D. Please understand us for this difficult. Thanks again.

-Line 238 – The authors state that H3K9me3 is enriched at the ALKBH5 promoter, but their argument is that this mark is being removed. Is this simply residual and it was previously higher? This is confusing

Response: Thanks for your question. As shown by figure 5B, in the si-NC group, the enrichment of H3K9me3 on the ALKBH5 promoter was higher than the negative control IgG. Therefore, we written that H3K9me3 was enriched on the ALKBH5 promoter. Thereafter, in the whole H3K9me3 enrichment, relative to the si-NC group, the enrichment of H3K9me3 on the ALKBH5 promoter was increased in the si-MIR205HG group, while relative to the si-MIR205HG + NC group, the enrichment of H3K9me3 on the ALKBH5 promoter was decreased in the si-MIR205HG + JMJD2C group. We have restated descriptions of statistical analysis and results to avoid confusion. Please review it again. Thanks.

-Lines 253 and 507– The phrase “stable low expression” is confusing. Consider “MIR205HG knockdown”.

Response: We have revised the expression as you suggested. Thanks for your help.

-Figure 7D going over two lines is awkward. Authors may wish to consider splitting it into three individual panels D-F and relabeling 5E as 5G.

Response: Thanks for your suggestion. We have rearranged pictures in figure 7. Please review it again. Thanks.

Discussion:

-Lines 264 and 316 – Please break up these sentences into smaller parts. Excellent use of Fig. 8 however.

Response: We have revised this sentence according to your suggestion. Thanks.

-Line 293 – Similar to the note above re Line 238, the statement that H3K9me3 is enriched is confusing and seemingly contradicted in the next sentence. Please clarify.

Response: Thanks for your suggestion. As we explained before, we revised the descriptions in the results section. Therefore, we also revised corresponding descriptions in discussion to avoid confusion. 

-Line 307 – Why was ALKBH5 protein level not analyzed? If possible/practical, including these data would add to the evidence already described.

Response: Thanks for your review and reading. Histone methylation is a process through which methyl groups are transferred to amino acids that make up the histones of nucleosomes, and the DNA double helix wraps around the nucleosomes to form chromosomes. Histone methylation can regulate gene transcription. JMJD2C is required for histone modification. Therefore, we only detected the transcriptional levels of ALKBH5 in this study. In the future, we will explore the changes in ALKBH5 protein levels and the specific mechanism that regulates ALKBH5 protein levels. Please keep an eye on our future study. Thank you.

Figure Legends:

-Line 459 – What is shown is not transfection efficiency but knockdown efficiency.

Response: We have revised the expression according to your suggestion. Thank you.

-Are CCK-8 assays done with both biological and technical replicates?

Response: In our study, the CCK-8 assays include three technical replicates and three biological replicates. We have stated it in the methods section. Thanks for your review.

-Figure 4 and 6 titles – “by silencing MIR205HG” at the end of the sentence is misleading. Please revise.

Response: We have revised the titles to avoid the misleading. Please review them again. Thanks.

-Line 513 – the name of the statistical analysis used was omitted. Please revise.

Response: We are sorry for that mistake. We have added the name of the statistical analysis here. Thanks.

Reviewer #2: The study is generally interesting and performed in a logical thought. However, it could be further improved from the following aspects.

1. In many places such as Line 243-244 (JMJD2C reduced H3K9me3 enrichment in the ALKBH5 promoter 244 region and promoted ALKBH5 transcription.), the tone should be softer since the conclusion was just based on the correlation analysis and literature. The direct effect and relationship among JMJD2C, H3K9me3 and ALKBH5 were not proved by the experiments.

Response: Thanks for your suggestion. We have revised relevant description according to your suggestion. Thanks for your help.

2. The language needs to be polished by some native speaker.

Response: We have made extensive revisions of the language. We hope that our revised language may satisfy you.

3. More original data such as CHIP, RIP should be provided as supplementary materials in the manuscript.

Response: Thanks for your review and question. We have provided data of ChIP and RIP in the form of supplementary materials. Please review it again. Thank you.

6.PLOS authors have the option to publish the peer review history of their article (what does this mean?). If published, this will include your full peer review and any attached files.

Response: Thanks for your inquiry. I choose yes.

Do you want your identity to be public for this peer review? For information about this choice, including consent withdrawal, please see our Privacy Policy.

Reviewer #1: Yes: Cai M. Roberts

Reviewer #2: No

Thank you for your careful review. We really appreciate your efforts in reviewing our manuscript. Your careful review has helped to make our study clearer and more comprehensive. Thank you again.

Best regards,

Lu Yan

michelley8051@126.com

---

## [Decision Letter · Decision Letter 1]

21 Aug 2023

The promotive role of lncRNA MIR205HG in proliferation, invasion, and migration of melanoma cells via the JMJD2C/ALKBH5 axis

PONE-D-23-08794R1

Dear Dr. Yan,

We’re pleased to inform you that your manuscript has been judged scientifically suitable for publication and will be formally accepted for publication once it meets all outstanding technical requirements.

Kind regards,

Xiangning Zhang, M.D., Ph.D.

Academic Editor

PLOS ONE

Additional Editor Comments (optional):

Re.: The promotive role of lncRNA MIR205HG in proliferation, invasion, and migration of melanoma cells via the JMJD2C/ALKBH5 axis, by Lu Yan et al.,

Dear Dr. Yan,

Thank you for submitting us your above-referenced manuscript for publication consideration. After two rounds of review, the questions raised during the peer reviews have been adequately responded, and the quality of it has attained to the level of publication. Currently, required number of reviews on the revision has arrived. After rexamining according to the criteria of publication proposed by Plos, a recommendation of acceptance is therefore advised to the editorial team.

Should you have any question regarding your manuscript, please feel free in contacting me.

Best,

Xiangning Zhang, M.D., Ph.D.

Academic Editor.

Reviewers' comments:

Reviewer's Responses to Questions

**Comments to the Author**

1. If the authors have adequately addressed your comments raised in a previous round of review and you feel that this manuscript is now acceptable for publication, you may indicate that here to bypass the “Comments to the Author” section, enter your conflict of interest statement in the “Confidential to Editor” section, and submit your "Accept" recommendation.

Reviewer #1: All comments have been addressed

Reviewer #2: All comments have been addressed

2. Is the manuscript technically sound, and do the data support the conclusions?

Reviewer #1: Yes

Reviewer #2: Yes

3. Has the statistical analysis been performed appropriately and rigorously? 

Reviewer #1: Yes

Reviewer #2: Yes

4. Have the authors made all data underlying the findings in their manuscript fully available?

Reviewer #1: Yes

Reviewer #2: Yes

5. Is the manuscript presented in an intelligible fashion and written in standard English?

Reviewer #1: Yes

Reviewer #2: Yes

6. Review Comments to the Author

Reviewer #1: (No Response)

Reviewer #2: (No Response)

7. PLOS authors have the option to publish the peer review history of their article (what does this mean?). If published, this will include your full peer review and any attached files.

Reviewer #1: **Yes: **Cai M. Roberts

Reviewer #2: No

---

## [Editor Report · Acceptance letter]

1 Sep 2023

PONE-D-23-08794R1 

The promotive role of lncRNA MIR205HG in proliferation, invasion, and migration of melanoma cells via the JMJD2C/ALKBH5 axis 

Dear Dr. Yan:

I'm pleased to inform you that your manuscript has been deemed suitable for publication in PLOS ONE. Congratulations! Your manuscript is now with our production department. 

Kind regards, 

on behalf of

Dr. Xiangning Zhang 

Academic Editor

PLOS ONE